# Estimation of Fluor Emission Spectrum through Digital Photo Image Analysis with a Water-based Liquid Scintillator

**DOI:** 10.3390/s21248483

**Published:** 2021-12-20

**Authors:** Ji-Won Choi, Ji-Young Choi, Kyung-Kwang Joo

**Affiliations:** Department of Physics, Institute for Universe & Elementary Particles, Chonnam National University Yongbong-ro 77, Puk-gu, Gwangju 61186, Korea; wldnjs707@naver.com

**Keywords:** liquid scintillator, water-based liquid scintillator, commission internationale de l’eclairage (CIE) color space, red green blue (RGB), hue saturation value (HSV), digital camera, complementary metal oxide semiconductor (CMOS), bayer color filter array (CFA), image analysis, sinogram, radon transformation

## Abstract

In this paper, we performed a feasibility study of using a water-based liquid scintillator (WbLS) for conducting imaging analysis with a digital camera. The liquid scintillator (LS) dissolves a scintillating fluor in an organic base solvent to emit light. We synthesized a liquid scintillator using water as a solvent. In a WbLS, a suitable surfactant is needed to mix water and oil together. As an application of the WbLS, we introduced a digital photo image analysis in color space. A demosaicing process to reconstruct and decode color is briefly described. We were able to estimate the emission spectrum of the fluor dissolved in the WbLS by analyzing the pixel information stored in the digital image. This technique provides the potential to estimate fluor components in the visible region without using an expensive spectrophotometer. In addition, sinogram analysis was performed with Radon transformation to reconstruct transverse images with longitudinal photo images of the WbLS sample.

## 1. Introduction

A conventional LS is a mixture of a base solvent and fluor [1,2,3,4,5]. In general, for a base solvent, an organic solvent is used. A scintillating fluor is a chemical material that absorbs photons and re-emits them at a longer wavelength. Its purpose is to pick up excitation energy from the solvent and emit a fraction of its energy as visible light. This emitted light is read by a photomultiplier tube (PMT). In addition, a secondary wavelength shifter (WLS) can be added to match the maximum quantum efficiency of the bi-alkali PMT around 400−450 nm [6]. In our study, 2,5-diphenyloxazole (C_15_H_11_NO, PPO) was used as a primary fluor, and 1,4-bis (5-phenyl-2-oxazolyl) benzene (C_24_H_16_N_2_O_2_, POPOP), or 1,4-bis (2-methylstyryl) benzene (C_24_H_22_, bis-MSB) were added as the secondary fluor. Organic LS has been widely used in the nuclear, particle, and medical physics fields, since it has proper optical properties and a low energy threshold. In general, the amount of optimized primary fluor in organic LS used in reactor neutrino experiments is approximately ~3 g/L, while the secondary WLS is ~30 mg/L [3,7]. In our study, when synthesizing a water-based liquid scintillator (WbLS), the same fluor concentration used in the reactor neutrino experiments was maintained.

Even though conventional LS has many advantages, WbLS has been proposed and studied as a future next generation detector [8,9,10,11,12]. It has a lower cost, is less hazardous, is more environmentally friendly, and has a longer attenuation length. In order to synthesize a liquid scintillator using water, it is essential to dissolve the organic liquid scintillator into water. The water and organic solvent do not mix with one another, and will quickly separate into two layers due to differences in polarity. A surfactant can be used to reduce the tension between the polar (hydrophilic) and non-polar (lipophilic) surfaces. In general, fluor dissolves in organic solvent, but does not dissolve directly in water. Therefore, with the help of a surfactant, fluor can be mixed with water and an organic solvent.

A surfactant consists of a polar hydrophilic group and a non-polar lipophilic group. The HLB index is one of the indicators used to characterize surfactants. This index is a measure of the degree to which it is hydrophilic or lipophilic, as described by Griffin [13,14] and Davies [15]. The HLB number usually ranges from 0 to 20. A low HLB value means that it has a lipophilic tendency to be more soluble in oil and to form water-in-oil emulsions, whereas a higher HLB value corresponds to a hydrophilic molecule. For this study, polyoxyethylene nonylphenylether (IGEPAL CO-630, (C_2_H_4_O)_n_, C_15_H_24_O, n = 9~10) was used to estimate the emission spectrum of fluor components in the WbLS. Its HLB value is 13 and it has a hydrophilic tendency. In sinogram analysis, PEG-60 hydrogenated castor oil (HCO-60, C_18_H_37_NO_3_) with a HLB value of 14 was used. If we know the HLB index value, a gel-type WbLS can be synthesized more easily. The original state of HCO-60 is a hard paste. If the ratio of water to surfactant exceeds (6:4), the viscosity of the WbLS sample becomes stronger. This sample was used for sinogram analysis later.

A high optical transparency for the WbLS is highly desired. A WbLS has the potential to provide a sufficient light yield with a long attenuation length and chemical stability over several years of experiments. Depending on the base organic solvent, types of fluor, fluor concentration, and surfactants used, our results of transmittance, absorption, fluorescence, and density were not significantly different from other previously measured values [9,11,16,17].

## 2. Motivation

There were two motivations for this paper. The first was to develop a new liquid scintillator using water as the main base solvent. Most WbLS research groups have mainly focused on finding a new surfactant, and then water and surfactant are mixed. But in our study, we synthesized a WbLS using only a combination of water, surfactant, and fluor according to the HLB index value. Secondly, we investigated the possibility to reconstruct the dissolved fluor components in a WbLS by analyzing photographic images taken with a digital camera (EOS 450D, manufacturer: Canon, Seoul, Korea) equipped with a CMOS (APS-C sensor, manufacturer: Sony, Seoul, Korea) image sensor. In general, an expensive UV/vis spectrometer and fluorescence spectrophotometer are used to obtain information on the absorption and emission spectrum of fluors. Our method will provide an inexpensive and indirect method to estimate fluor components in a sealed LS or gel-type materials. In addition, we tried to show that two-dimensional images could be reconstructed from projection data obtained from various directions of a WbLS sample. A tomographic image, or an axial cross-section image based on the reference line, was reconstructed by sinogram or Radon transformation.

## 3. Photo Color Image Processing Analysis

### 3.1. Color Filter Array (CFA) Image Sensor

Digital cameras equipped with CMOS technology were used for the digital image analysis. Each pixel of most commercial CMOS image sensors is equipped with a CFA. The CFA configuration in the CMOS is a Bayer filter mosaic consisting of red (R), green (G), and blue (B) filters that can cover a broad area of color space. In CFA, only one color among RGB is recorded at each pixel. The other missing two color values are estimated from the recorded mosaic data of RGB values through an interpolation process called demosaicing (or demosaicking) [18,19]. Numerous demosaicing algorithms have been proposed and among them Bayer CFA is widely used. The missing data for each color channel is estimated based on neighboring pixel information. On the contrary, the diffraction grating-based spectrometer has a different grating constant according to the refractive index, so it can form a fine baseline that can be distinguished at the level of 1 nm. Several integration technologies of CFA-based CMOS image sensors have been developed [20]. One workflow example is shown in Figure 1. The disadvantage of a CFA-based camera is that the original color is decomposed into three color filters and cannot be accurately expressed when converted back to the original color during the demosaicing process. In other words, the demosaicing process in Bayer CFA cannot represent the original color due to the lack of information when converting back to the original color. In addition, the effects of optical or electrical cross talk due to CMOS pixel structure cannot be ignored, so color correction is required [21]. Manufacturers apply an algorithm that optimizes the signal-to-noise ratio in order to achieve accurate color reproduction. In the neutrino experiment, there is no need to distinguish the wavelength of light entering the PMT down to a few nanometers. Therefore, we considered a method which could easily identify the fluor contents by analyzing the WbLS emission spectrum, and hence adopted the Bayer CFA approach for the demosaicing process.

### 3.2. Color Spaces

XYZ color space was defined by the Commission Internationale de l′Eclairage (CIE, International Commission on Illumination) in 1931 [22,23]. The CIE 1931 color space is used today as a standard to define colors, and as a reference for other color spaces. When (R, G, B) values are combined, the CIE model can reproduce almost any color that a human eye can perceive. However, the hue (H), saturation (S), value (V) model, an alternative representation of the RGB model, says that color is not defined as a simple combination of adding or subtracting primary colors, but that it is a non-linear mathematical transformation [24]. If RGB values are known, RGB can be converted to HSV values. Physically, hue is related to wavelength for spectral colors. Therefore, a wavelength can be obtained by a dominant hue value of the spectrum using the appropriate conversion method.

### 3.3. Emission Spectrum of Fluors from a Color-Decoded Image of WbLS

As an application, we investigated the possibility of determining the fluor components dissolved in the WbLS through a digital photo image analysis in color space after irradiating UV lights on the sample. For the background rejection, experiments were performed in a darkroom. Pictures were taken with the CMOS camera approximately 50 cm in front of the WbLS sample container. The camera was focused at the desired point by taking several pictures in advance, and efforts were made to place the camera as perpendicular as possible to the front of the sample plane. The angle between the camera and the WbLS sample container needed to be well aligned. By marking the coordinates of each point, we were able to position them as square to each other as possible. In addition, the refraction effect was examined. A WbLS sample was placed in the center of the camera view and photographed without magnification.

After a photo is taken with a digital camera, users can choose lossless raw data or lossy compressed data formats. In our case, we used a jpeg format corresponding to the second case. Only a certain number of pixels in the digital world were used through down-sampling, discrete Fourier integral transformation, and encoding processes. Then, R, G, and B values of each pixel were stored in a color look-up table with a scale of 256, since we were using an 8-bit digital camera.

A schematic diagram for taking digital image photographs is shown in Figure 2a. To prevent external background lights from entering the camera lens, a wall between the camera and the WbLS sample was installed. The digital camera was positioned at a 90° angle to the axis of the beam, so the light emitted from the UV lamp did not enter the camera directly. In addition, we tried to prevent any stray lights from entering the camera. Figure 2b shows a light image emitted from the WbLS sample using IGEPAL CO-630 surfactant. These images were taken with a digital camera using a few seconds exposure time. Only those pixels whose V value in the HSV model was greater than roughly 60% were selected to remove background. This boundary line was indicated as a rectangular box. The fourth box was selected for the analysis. By using a desktop computer with a CPU (Ryzen 7 3700X, manufacturer: AMD, Seoul, Korea), GPU (Radeon RX5700, manufacturer: AMD, Seoul, Korea), and 32 GByte of RAM (DDR4-3200, manufacturer: Samsung Electronics Co., Ltd., Seoul, Korea) the analysis of the stored image can be performed within several minutes.

Figure 3a−c shows each (R, G, B) component distribution in color space as a function of the color intensity value. The pixel intensity value was obtained from the look-up table with a 256 value scale. Our WbLS contains three fluor substances that convert from UV to visible light. Those were PPO, POPOP, and bis-MSB. Emission peaks of PPO, POPOP, and bis-MSB lie near wavelengths 360, 410, and 420 nm, respectively [6,25,26]. The blue side light intensity is greater compared to the red or green pixel intensity value. Extracted emission spectrum of PPO, PPO+POPOP, and PPO+bis-MSB from each hue value after background subtraction is shown in Figure 3d) as a function of wavelength. That the emission peak of PPO appears around 380 nm rather than 360 nm is due to the intrinsic limitations of the CMOS Bayer CFA configuration. The camera we were using is not sensitive below 375 nm and does not reconstruct colors properly [27]. Therefore, special care and attention were required in this wavelength region. The emission spectrum difference between POPOP and bis-MSB is about 10 nm. We could clearly distinguish the difference of emission spectrum between PPO, POPOP, and bis-MSB fluors especially in the blue-like color region. This method has sufficient potential to estimate emission spectrum in the visible region without opening or extracting samples from a sealed liquid scintillation detector.

### 3.4. Sinogram and Optical Tomography

As a second example, we examined the sinogram [28]. This is a visual representation of the raw data by taking photos from different angles. In 1917, Johann Radon demonstrated the theory that two-dimensional images could be reconstructed from projection data taken from multiple directions of an object. The accumulated value of light transmission passing through the object is displayed from 0° to 180° (or 360°) on the horizontal axis, and the height of the graph for each angle is visualized using contrast. Mathematically, the sinogram is a result of performing the radon transformation. In order to get the original image, inverse Radon transformation is performed.

A WbLS sample for sinogram analysis is shown in Figure 4. Gel-type WbLS using HCO-60 surfactant was synthesized. We can easily make and check non-homogeneity with a gel-type WbLS. This gel-type WbLS has the advantage that an air bubble can be created in a specific part of the sample. Figure 4a shows the procedure of the sinogram. UV lights with wavelength of 250, 310, and 360 nm were illuminated from the top to the bottom of the WbLS sample. The disk on which the WbLS sample was placed rotated at a constant speed. The digital camera and rotating disk were operated by a drive control module. A digital photo was taken after rotating the disk every 0.25°.

Because the rotation plate and camera were fixed, only the sample rotated about the rotational axis without moving up and down in the front camera view. The z-axis height of the sample and the z-axis height of the acquired image were the same. The camera was focused on the horizontal dashed line passing through the center of a small tube embedded in the WbLS sample. The dashed line is the baseline used to reconstruct the tomography image, and the z-axis pixel number is 542. The image was extracted from this line. A total of 1.400 photos were taken, and only a fraction of them are shown in Figure 4b.

Figure 5a shows a 360° sinogram reconstructed from images with the air bubbles in WbLS. The air bubble was formed as a hard gel, and the size of the bubble was relatively small, so the structural shape of the gel was maintained for a long time. As already mentioned in Figure 4a, when looking at the sample container from top to bottom, and assuming Cartesian coordinates, the air bubble was created at the center of WbLS container. In addition, the tomographic top view image can be reconstructed after inverse Radon transformation. When viewed from the top down, Figure 5b provides the information that there is an object (two air bubbles) at the central region of the WbLS sample container. This shows that we can accurately reconstruct the position of the air bubbles we created. Therefore, from this result we can acquire a cross-sectional view image with a longitudinal photo image taken while rotating the WbLS sample. The ring shape of the WbLS sample container was also visible. Unlike the X-ray used in computed tomography (CT), the optical tomography technique we attempted here was the visible light. Due to diffraction and reflection between photons, the wave properties become more prominent. If diffraction and reflection are very severe, parallel beam attenuation assumptions cannot be satisfied, and then ill-posed inverse problems result in ghost (or artifact) images in the final reconstructed image. Therefore, the air bubble was reconstructed as a distorted or slightly deformed circle rather than a perfect circle due to its high diffraction and high reflectivity.

## 4. Summary

We synthesized a WbLS based on the HLB index value of surfactant. To date, most WbLS research groups have mainly focused on finding a new surfactant, and then water, organic solvent, flours, and surfactant are mixed. In our case, water, fluors, and surfactant are mixed based on the HLB index. Two surfactants (IGEPAL CO-630, HCO-60) were synthesized for the WbLS. IGEPAL CO-630 was used to estimate the emission spectrum of the fluor dissolved in the WbLS. HCO-60 was used for sinogram analysis with a gel-type WbLS. By applying a combination of WbLS and CMOS sensor, it was possible to perform imaging analysis related to the fluor component of the WbLS. Photo images of UV lights onto the WbLS sample were taken by a CMOS digital camera. CMOS image sensors with Bayer CFA and a demosaicing process were used to reconstruct and decode color. By analyzing these images, the possibility to obtain the flour component dissolved in the WbLS was investigated. This method provides a simple technique to identify the emission spectrum of fluors in the WbLS, even with a mobile phone camera. For a second example, a rotating measurement system for sinogram analysis was implemented. A gel-type WbLS was prepared and an air bubble was created in the sample. It was possible to reconstruct transverse images with longitudinal photo images of the WbLS sample through Radon transformation.

## Figures and Tables

**Figure 1 sensors-21-08483-f001:**
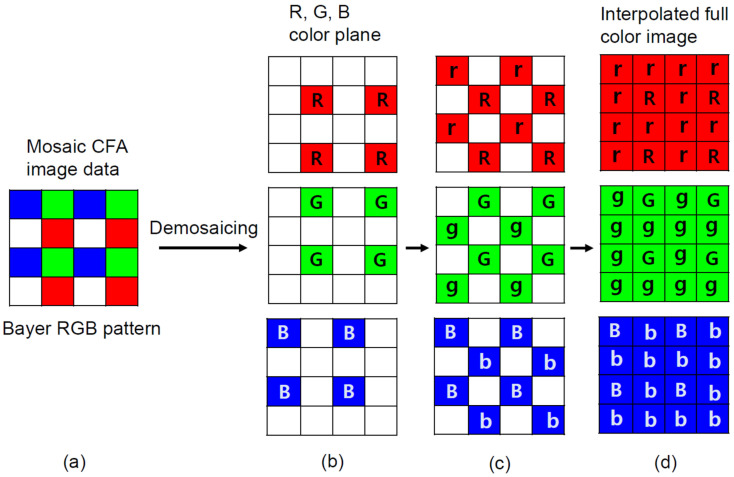
A workflow example of CFA arrangement and the demosaicing process for an interpolated full color image. (**a**) Repetitive patterns consisting of R, G, and B are called mosaics. (**b**) Demosaicing according to each RGB color component. Input raw (mosaiced) image from CFA. (**c**) Spatial arrangement (lowercase of rgb) of each pixel is assigned based on neighboring pixel color information. The result of red, green and blue channel interpolation at white locations. (**d**) The final result of RGB interpolation of each pixel. The final result of red, green, and blue channel interpolation at red, green and blue locations.

**Figure 2 sensors-21-08483-f002:**
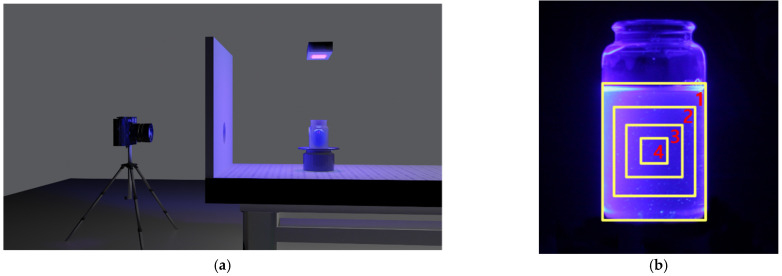
(**a**) Experimental setting for taking digital image photographs. The camera was placed approximately 50 cm in front of the WbLS sample. Because there was a wall between the camera and the WbLS sample the UV lamp light did not directly enter the camera. Only the desired light reached the camera. A cylindrical quartz container with a diameter of 4 cm and a height of 7 cm was filled with WbLS using IGEPAL CO-630 surfactant. It was placed on top of the rotating disk. UV light illuminated the sample from the top of the container. The camera was remotely controlled. (**b**) Rectangular boxes represent regions of interest. Only those pixel regions whose V value in the HSV model was greater than 60% were selected and their boundary lines were displayed as a rectangular box. The fourth box was selected for the background rejection.

**Figure 3 sensors-21-08483-f003:**
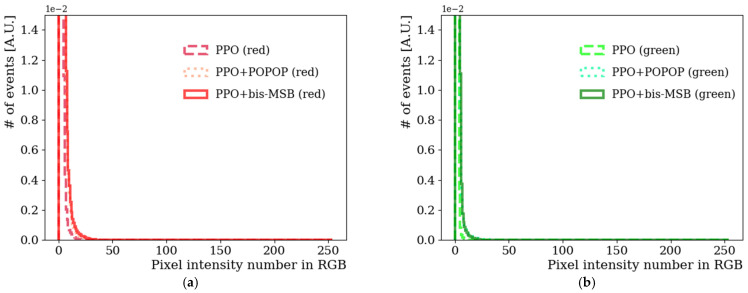
(**a**) Red, (**b**) Green, and (**c**) Blue components extracted from the photographed images in Figure 2b taken by a CMOS digital camera (Canon EOS 450D). RGB values of PPO, PPO+POPOP, PPO+bis-MSB as a function of pixel intensity are listed. Among R, G, and B values, blue values are dominant in each fluor case. (**d**) Extracted emission spectrum of PPO, POPOP, and bis-MSB from hue value after background subtraction as a function of wavelength.

**Figure 4 sensors-21-08483-f004:**
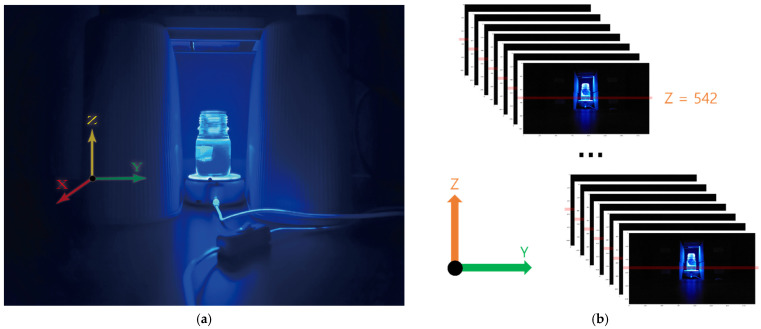
(**a**) UV light was illuminated from above the WbLS sample. An initial front view digital image of the WbLS sample using HCO-60 surfactant placed on a rotating disk before rotation. Then, the disk rotated at a constant speed. (**b**) A horizontal dashed line is shown passing through the air bubble in the container. Its pixel number of z-axis is 542. The tomographic image was extracted and reconstructed based on this line. The camera was focused on the center of the container and a picture was taken after rotating the disk every 0.25°. Only a portion of the 1,400 projected images is shown.

**Figure 5 sensors-21-08483-f005:**
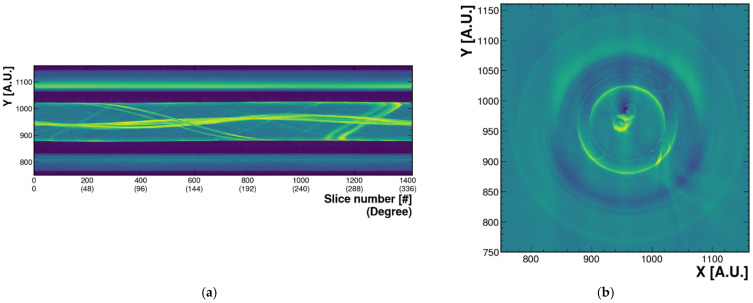
(**a**) A 360° sinogram reconstructed from the image of the small air bubble created in WbLS sample. The x-axis is 1 to 360° (1 to 1400 pixel labels), depending on how much the cylindrical circle is divided. The y-axis is the pixel label of the y-axis (height) of the yz-plane image. The color represents the brightness of the surroundings including the bottle when we reconstruct it with an arbitrary color. Each pixel number band on the y-axis means 700–800 (outer wall), 800–900 (air layer between sample container and outer wall), 900–1000 (inside WbLS sample container), 1000–1100 (air layer between sample container and outer wall, same as 800–900 pixel numbers), 1100–1200 (outer wall, same as 700–800 pixel numbers). (**b**) Tomographic top view image after inverse Radon transformation from (**a**). The pixel number of z-axis is 542. The x(y)-axis is the pixel label of the x(y) coordinate seen from the top. Two air bubbles in the center of the WbLS sample container can be seen. The ring shape surrounding the air bubbles represents the container. The outermost circle-shaped image represents a virtual image created by light reflected from the outside of the WbLS sample container.

## Data Availability

Not applicable.

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
