# Peer review of "Estimation of Fluor Emission Spectrum through Digital Photo Image Analysis with a Water-Based Liquid Scintillator"

_sensors, 2021, doi:10.3390/s21248483_

Round 1

Reviewer 1 Report

The paper explores the analysis of determining the fluor component and optical tomography using digital color processing. An interesting concept from instrumentation perspective. Usage and application can be expanded. 

Please add line numbers on manuscript. It’s difficult to make comments.

General comments:

Several questions in results, particularly on On page-4: “Figure 3 (a)~(c) shows the distribution of each (R, G, B) component in color space as a function of the color intensity value. Pixel intensity value was obtained from the look-up table with a 256 value scale. Our WbLS contains three fluors substances that convert from UV to visible light, PPO and POPOP or bis-MSB. PPO emits photons at 300 ~ 500 nm and its peak value is approximately 380 nm [22]. POPOP and bis-MSB emit photons at 380 ~ 550 nm and their peak wavelength are around 410 nm and 430 nm [23], so the blue side light is relatively larger compared to the red or green pixel intensity value.”

PPO is a common fluor that has been widely used in scintillator cocktails and scintillator-based neutrino experiments over the past decades. It has well-known (and been extensively studied) absorption and emission regions/peak-max. There are many references and webpages for published data.

Q. please refine the statement of “PPO emission at 300~500nm” which is very vague. Figure 3d only presented a rough region from 375~400nm; and I cannot find one from ref [22] as cited. in the statement

Q. it’s unlikely that PPO emits fluorescence at close to 300nm which is shorter than its absorption wavelength.

Q. Figure 3d showed PPO emission peak at 380nm, which is 20nm away from the known value of ~360nm. After reviewing reference [22], there is also no indication to support the 380nm as PPO emission peak from this work. Please comment.

Q. “POPOP and bis-MSB emit photons at 380 ~ 550 nm and their peak wavelength are around 410 nm and 430 nm [23], so the blue side light is relatively larger compared to the red or green pixel intensity value.”

These sentences are not clear, particularly the later part. What is the correlation?

Q. What process was used to convert the pixel intensity number to wavelength in Figure 3 (i.e. 3c to 3d)? How do you reconstruct and calibrate the wavelength position? and what is resolution? recommend to add a section or sentences to clarify the reconstruction or calibration.

Q. What is the uncertainty for Photo color image processing analysis? For multiple photos, have there been reproducibility studies, repeating measurements after sample disassembling and reassembling the setup? Please add a section or few sentences to explain uncertainty/errors.

Q. Could you expand on at what occasions of the non-invasive method determining the fluor component dissolved in scintillator detectors? In which situations would it be useful?

Q. The Summary section is not well organized with some questionable sentences. It didn’t conclude much of what have been accomplished from this work, but repeating those have been described in introduction/experiment sections already. The sentences read not conclusive (possibly, possibility, etc.) and lack of correlation. Recommend to rephrase and reorganize this section.

Examples of questions: (no line number to refer)

Q. “We have developed scintillating liquids using water as a solvent for an organic liquid scintillator…”  What does it mean using water for an organic liquid scintillator? These are two different categories of liquids.

Q. “Two surfactants (IGEPAL CO-630, HCO-60) were synthesized for WbLS…”

Q. Did the authors synthesize these surfactants for use in WbLS?

thanks.

“Overall, we hope that WbLS and our application could be used in the future, next generation particle detectors or other related fields”

Which experiments with any specific usages? the color imaging analysis or WbLS (which doesn’t seem to be the center of this work)?

Suggest to add some useful references of WbLS applications in neutrino physics from THEIA or other experiments.

Author Response

Please take a look at an attached file. Thanks. 

Reviewer 2 Report

The paper covers the analysis of determining the fluor component in WbLS and optical tomography using digital color processing. I recommend some revisions that would make the motivation and conclusion of the paper more evident and complete before the publication. 

Abstract 

Abstract need sufficient work. Only 
“We can estimate the emission spectrum of the fluor dissolved in the WbLS by analyz-ing digital images. This technique has sufficient potential to estimate fluor components in the visible region without using an expensive spectrophotometer. In addition, sinogram analysis was per-formed with Radon transformation to reconstruct transverse images with longitudinal photo images of the WbLS sample. “

Seems to be the part appropriate for abstract. Introduction of WbLS requires a re-write, HLB and CMOS parts may not e relevant for abstract, motivation of work should be included. 

Introduction 

- Line 3: re-emit -> re-emits 
- Line 4: light -> visible light
- Last sentence of the first paragraph: WbLS abbreviation hasn’t been introduced in the main text yet. 
- First sentence of the second paragraph: LS using water -> WbLS 

- Explore the motivation for moving to WbLS (cost, environmentally-friendly, attenuation length). 

Usage of “oil” throughout the text is confusing, for example Sentence “Therefore, with the help of surfactant, fluor can be mixed with water and oil. “ and then “we synthesized WbLS using no oil, “ 

The information of the last paragraph would probably be more organic to the reader after understanding the motivation of the paper.

Motivation
Can the digital camera method be used for the quality control of the material throughout long periods of time? For example, years? Is there a possibility that flours fall out and their concentration is getting reduced in the sealed detector and you can notice with just pre-installed cameras? In other words, can you make a claim on concentration of the particular Fluor in comparison with just yes/no this Fluor is present in this sample.
Photo color image processing analysis 
How long does the analysis of the photos take? Is it computing resources intense?
Is my understanding correct that you used three samples, 1) PPO, 2) PPO+POPOP 3) PPO+bisMSB dissolved in water with IGEPAL CO-60 surfactant. Were these samples otherwise identical? What was the exact concentration of the flours in each of these samples? Please make it more clear throughout the text
- How does the Figure 3 d) compare with the emission spectra obtained with more conventional methods?

Sinogram and optical tomography 

What is the light source?
What is the contribution of WbLS? Would the same optical tomography setup work with any transparent gel?

Author Response

(The authors gave the same response as above.)

Reviewer 3 Report

The paper describes a new method to estimate composition for water based liquid scintillator (WbLS).
This method use only CMOS sensor with Bayer color filter array (CFA) and thus it could provide a low-cost experimental setup.
The result is not very impressive but maybe useful in some areas.

They also show the result of Sinogram analysis that provide photo images of WbLS. 
However, I do not see the motivation and the novelty of this approach. 

Overall, I think the quality of the paper is at a publishable level.
Here are several comments and questions listed below.

1. Introduction 
In the introduction chapter, you explain organic LS is used in nuclear, particle and medical physics field.
I wonder what is the motivation of WbLS development. What kind of application do you aim by WbLS? You need explain.

2. Motivation
Here, you claim that first motivation of this paper is new WbLS development, but this is not reported or discussed in this paper.
 How you determine the components using HLB value?
 Does HLB value important in this paper? It is not mentioned at all. You may need to add subsection such as "New WbLS development".
 What was the result of new WbLS development from the view of light yield and transparency of WbLS?

Figure 1
As far as I know, Bayer color filter include (R:G:B) with a fraction of (1:2:1). Am I wrong?
You need to explain the difference between uppercase and lowercase letters of the alphabet.
Also, you better explain what happens in the process (b)->(c), (c)->(d) respectively.

Figure 2
Additional information such as distance, size etc. should be included in the figure 2-(a) or caption.
In figure 2-(b), what do the region 1, 2, 3, 4 represent? How you use them in the analysis?

Figure 3
Emission wavelength difference between PPO+POPOP and PPO+bis-MSB is 20 nm according to the text, but it seems about 10 nm in the figure 3-(d).
You need to explain this result, and also you better explain how you calibrate X-axis of figure 3-(d).

3.4. Sinogram and optical tomography.
What is the motivation and novelty of this measurement?
Also, I could not understand what "the central axis for each angle" means.
How long does it take for completing this measurement? Doesn't the bubble move during the measurement?

Figure 5
Label of each figure should be (a) and (b), not (c) and (d), aren't them?
What is z-axis (color) of these figure?
If possible you better to write what each line represents on the diagram.

4. Summary
You say WbLS and your application could be used in next generation particle detector.
You should discuss how these results will help the particle detectors.
This result is new in that it suggests a way to investigate the composition of WbLS, how will it help the next generation of particle detectors?

Author Response

(The authors gave the same response as above.)

Round 2

Reviewer 1 Report

Thanks for the undated version. The main question remains as the data interpretation and results. The question of emission max repeated below:

Q. Figure 3d showed PPO emission peak at 380nm, which is 20nm away from the known value of ~360nm.

In your responses 

(1) described in cover letter: "References were replaced with [24, 25, 26]. Emission information of PPO/POPOP/bis-MSB can be found at Table 1 in reference [25]".

(2) described in revised version of lines 186-187: “Emission peaks of PPO, POPOP, and bis-MSB lie near wavelengths 360, 410, and 420 nm, respectively [24, 25, 26].”

Exactly this is the question that all referenced values (including tabe-1 in Ref [25]) for emission max of PPO is 360nm, but the present value shown in Figure 3d, the PPO emission max is 380nm (which is a difference of 20nm). In addition the present PPO emission spectrum does not even cover 360nm or agree with referenced spectrum of 340-400nm. Is this due to calibration or error assignment? Neither sections 3.2 nor 3.3, as the authors' responses, address why the present results didn’t agree with that from references. 

Author Response

Thanks for your valuable comments. Please take a look at an attached file. 

Thanks. 

Reviewer 3 Report

The paper describes a new method to estimate composition for water based liquid scintillator (WbLS). This method use only CMOS sensor with Bayer color filter array (CFA) and thus it could provide a low-cost experimental setup. The result could be useful in some areas.They also show the result of Sinogram analysis that provide photo images of WbLS.

The new manuscript is well modified from the last version of this manuscript. Overall, I think the quality of the paper is at a publishable level. 

Author Response

One more time, thanks for your valuable comments.
